# Interplay Between Vascular Dysfunction and Neurodegenerative Pathology: New Insights into Molecular Mechanisms and Management

**DOI:** 10.3390/biom15050712

**Published:** 2025-05-13

**Authors:** Avanthika Mekala, Hongyu Qiu

**Affiliations:** 1Cardiovascular Translational Research Center, Department of Internal Medicine, College of Medicine-Phoenix, University of Arizona, Phoenix, AZ 85004, USA; amekala@arizona.edu; 2Clinical Translational Sciences (CTS) and Bio5 Institution, University of Arizona, Tucson, AZ 85721, USA

**Keywords:** aging, vascular pathology, dementia, Alzheimer’s disease (AD), neurodegenerative disorder

## Abstract

Vascular dysfunction frequently coexists with neurodegenerative disorders such as dementia and Alzheimer’s disease (AD) in older individuals; however, the cause-and-effect relationship remains unclear. While AD is primarily characterized by neural tissue degeneration, emerging evidence suggests that aging-induced vascular dysfunction contributes to both the onset and progression of cognitive impairment and dementia by decreasing cerebral blood flow (CBF) and disrupting the blood–brain barrier (BBB). This challenges the traditional notion and underscores vascular dysfunction as an early pathogenic stimulus; thus, targeting vascular pathologies could be a promising strategy to slow dementia progression and potentially prevent AD. Conversely, aging-related neurodegeneration exacerbates vascular dysfunction, accelerating dementia pathology through oxidative stress and inflammation as well as deposition of neurotoxic substances such as beta-amyloid (Aβ) and tau in vascular walls. This bidirectional interaction creates a vicious cycle that worsens cognitive decline, underscoring the complexity of these diseases. This review aims to highlight recent advances in research on the mechanisms of aging-related vascular dysfunction in neurodegenerative diseases, focusing on vascular contributions to cognitive impairment and dementia (VCID) and AD. Additionally, we will explore the reciprocal effects and intricate relationship between vascular dysfunction and neurodegenerative pathologies, enhancing our understanding of relative disease pathogenesis and guiding the development of innovative prevention and treatment strategies.

## 1. Introduction

The risk of developing cognitive impairment and dementia increases significantly with age, doubling approximately every 5 years. Currently, there is no cure, making these disorders a growing public health concern, particularly as life expectancy continues to rise. Alzheimer’s disease (AD) is the most common form of dementia, accounting for 60–80% of cases. It is a neurodegenerative disorder characterized by the accumulation of beta-amyloid (Aβ) and phosphorylated tau in brain tissue, leading to progressive neuronal damage and synaptic loss. Traditionally, AD has been viewed as a neuron-centric disorder featuring amyloid plaques, neurofibrillary tangles, and brain atrophy. However, emerging evidence suggests that vascular dysfunction may precede or accompany neurodegeneration, serving as an early pathogenic trigger [1,2,3,4,5]. This vascular hypothesis of AD underscores the critical role of vascular impairment in the development and progression of cognitive decline.

With age, both the structure and function of the vascular system undergo significant changes, affecting not only the cerebral vasculature but also systemic circulation. Given the vascular system’s essential role in oxygen and nutrient delivery, waste clearance, and immune regulation, its dysfunction, marked by reduced blood flow, endothelial damage, and chronic inflammation, has been increasingly recognized as a key contributor to neurodegenerative diseases. Vascular dysfunction impairs cerebral perfusion and compromises the integrity of the blood–brain barrier (BBB), thereby facilitating neurodegenerative processes.

Distinguished from AD, vascular contributions to cognitive impairment and dementia (VCID) is the second most common cause of dementia [6,7,8], arising from conditions causing clear vascular damage or dysfunction, such as small vessel disease, chronic hypoperfusion, and stroke. These lead to ischemia, microinfarcts, white matter damage, and other cerebrovascular abnormalities, including lacunar infarcts and microbleeds. VCID is often characterized by an aging neurovascular unit (NVU) that becomes increasingly vulnerable to systemic insults such as metabolic disorders, inflammation, and proteinopathies. While AD is hallmarked by memory impairment due to neuronal degeneration, VCID more commonly presents with executive dysfunction and slowed cognitive processing caused by vascular injury.

Although AD and VCID have distinct etiologies and pathological features, they frequently coexist, contributing to mixed dementia and complicating clinical diagnosis and treatment. Importantly, age-related vascular changes, including arterial stiffening, BBB breakdown, and reduced cerebral perfusion, contribute to both VCID and AD pathology. Conversely, AD-related neurodegeneration can worsen vascular function through increased oxidative stress, inflammation, and Aβ deposition in the vascular wall [9]. This bidirectional relationship creates a vicious cycle in which vascular dysfunction and neurodegeneration reinforce one another, accelerating cognitive decline. Understanding this complex interplay between vascular health and neurodegeneration has profound implications for disease pathogenesis and clinical management.

By integrating insights from recent research, this review aims to summarize the evolving understanding of vascular contributions to neurodegenerative diseases, highlight potential therapeutic targets within the vascular system, and examine emerging strategies for prevention and treatment. Comprehensive knowledge of vascular involvement in neurodegenerative diseases is essential for developing innovative therapeutic approaches. Early detection and effective management of vascular risk factors may help prevent or delay the onset of cognitive impairment. Ultimately, addressing vascular dysfunction offers a promising avenue to combat dementia, improve clinical outcomes, and mitigate the growing public health burden posed by aging-related cognitive decline.

## 2. Vascular Aging and Its Impact on Cognitive Decline and Dementia

AD and VCID are the two major causes of dementia, but they differ in their underlying mechanisms, risk factors, and pathological changes. While AD is primarily known as a neurodegenerative pathology, VCID is caused by vascular issues, such as large or small vessel diseases that reduce blood flow to the brain or strokes, leading to chronic or acute ischemia, microinfarcts, and white matter changes. Aging-related vascular changes significantly contribute to both AD and VCID.

### 2.1. Aging-Induced Systemic and Cerebral Vascular Dysfunction

Aging induces systemic vascular structural alterations, particularly in the ascending aorta, peripheral arteries of the upper and lower limbs, and vertebral and basilar arteries within the cerebrovascular system, which includes increased arterial wall thickness and enlarged arterial diameter [10,11,12,13,14,15,16]. These changes are linked to vascular cell proliferation/migration and extracellular matrix (ECM) remodeling, such as elastic degradation, lipid infiltration, and reduction in the elastin-to-collagen ratio [10,11,12,13,14]. These structural alterations contribute to various forms of vascular dysfunction. Arterial stiffening, for instance, reduces vascular elasticity and increases blood pressure [11,12,13,14]; endothelial dysfunction, due to decreased nitric oxide (NO) production, leads to impaired vasodilation [17,18]. Capillary rarefaction, characterized by reduced micro vessel density, limits oxygen and nutrient delivery to tissues [19,20]. Collectively, these impairments reduce blood perfusion to vital organs, including the brain, ultimately affecting their function [10].

Multiple molecular and cellular alterations contribute to these age-related vascular changes. Vascular smooth muscle cells (VSMCs) experience phenotypic changes during aging, facilitating arterial stiffening and altered mechanical properties, which are associated with the activation of stiffness-associated signaling [14,21], phenotype shifting, dysregulated proliferation, and migration [22,23]. Vascular calcification, driven by inflammation, oxidative stress, mitochondrial dysfunction, and other external factors, also contributes to arterial stiffening and reduced distensibility [10,24]. Similarly, vascular endothelial cells undergo senescence, losing their proliferative ability and accumulating oxidative stress [10,25,26]. Aging is also associated with a decline in both the levels and regenerative capacity of endothelial progenitor cells, which are essential for maintaining endothelial function [10]. Furthermore, the adventitia that regulates vascular tone also increases with aging, which manifests as increased expression of total collagen in the artery [10].

These systemic vascular pathologies are mirrored in the cerebral vasculature. Both intracranial and extracranial arterial stiffening impair cerebral blood flow (CBF), raising the risk of stroke and transient ischemic attacks (TIAs), which may lead to neurodegenerative processes [6]. Endothelial dysfunction can compromise the integrity of the BBB, potentially allowing harmful substances to infiltrate the brain and disrupt neuronal function [27,28]. Furthermore, cerebral microvascular rarefaction and small vessel damage can cause chronic cerebral hypoperfusion, limiting oxygen and nutrient supply to brain tissue. The NVU, a complex of neurons, glial cells, and blood vessels, also becomes dysregulated with aging, impairing neurovascular coupling and exacerbating neurodegeneration [27,29].

These changes collectively contribute to the development of cerebrovascular diseases [10]. Given the essential role of the vascular system in delivering oxygen, nutrients, and immune cells, cerebral vascular dysfunction, characterized by impaired blood flow, endothelial damage, and BBB disruption, emerges as a key contributor to neurodegenerative diseases such as cognitive impairment, dementia, and AD. This highlights the critical importance of vascular mechanisms in these pathologies and underscores their potential as therapeutic targets.

### 2.2. The Role of Aging-Related Vascular Dysfunction in AD

AD is the major cause of age-related diseases and the fifth leading cause of death worldwide. The primary characteristic of AD is considered extracellular Aβ deposition in brain parenchyma, which is the most common cause of dementia [30]. Another main feature of AD is the accumulation of phosphorylated tau, which causes neuronal loss and glial activation from the formation of neurofibrillary tangles [30]. In addition, disruptions in the NVU involving endothelial cells and pericytes are also related to AD pathogenesis, demonstrated by reduced coverage of microvessels by pericytes, increased Aβ deposition, and reduced mRNA expressions of astrocytic end-feet water channel aquaporin [30]. Furthermore, cerebral amyloid angiopathy (CAA) is caused by the accumulation of the soluble monomeric Aβ 1–40, which is linked to white matter hyperintensities (WMH), cerebral hemorrhages, and weakened vascular walls [31].

Despite the precise mechanisms underlying AD, it remains incompletely understood. The primary pathology in AD is the increased Aβ deposition in the brain due to the impairment of Aβ clearance, which has been linked to dysfunction of Aβ-degrading enzymes in brain tissue [31]. Studies indicate that peripheral clearance plays a critical role in Aβ clearance, as 60% of brain-derived Aβ is transported from the BBB to the periphery to be cleared [31]. Thus, cerebrovascular insufficiency may contribute to the impairment of Aβ clearance. It has been shown that impaired vascular function reduces Aβ clearance via the glymphatic and perivascular systems [32,33,34].

BBB dysfunction allows harmful substances (e.g., inflammatory cytokines, toxins) to enter the brain, increasing neuroinflammation, exacerbating Aβ accumulation, and accelerating neurodegeneration. Aging-related chronic cerebral hypoperfusion reduces oxygen supply, causing subtle but cumulative ischemic damage. This contributes to white matter lesions and cognitive decline. These changes further reduce perfusion and impair waste clearance, promoting neurodegeneration.

### 2.3. Vascular Contributions to Cognitive Impairment and Dementia (VCID)

VCID represents the second most common form of dementia, accounting for approximately 20% of cases, with risk increasing significantly with age [6]. Pathological hallmarks of VCID include cerebral small vessel diseases (CSVDs), lacunar infarcts, and microbleeds [35,36]. According to the Vascular Impairment of Cognition Classification Consensus Study (VICCCS), VCID is categorized into four subtypes: post-stroke dementia, subcortical ischemic VCID, multiple-infarct (cortical) dementia, and mixed dementia [37].

VCID encompasses a diverse range of underlying vascular pathologies, including atherosclerosis, arteriosclerosis, infarcts, white matter changes, and microbleeds [6]. It is characterized by brain lesions that arise from vascular pathology, leading to ischemic, hemorrhagic, or hyperperfused states, each of which can contribute to a variety of clinical symptoms [38]. These symptoms vary depending on the severity and anatomical location of the injury [38]. Common clinical manifestations include pure motor or sensory deficits, sensorimotor symptoms, ataxic hemiparesis or gait disturbances, dysarthria, cognitive dysexecutive syndrome, and depression [39]. The spectrum of severity ranges from mild cognitive impairment to severe vascular dementia (VaD) [40,41]. A clearer understanding of VCID requires examining the various underlying mechanisms that contribute to its progression.

Despite the heterogeneity of VCID pathologies, a central feature is chronic and significant cerebral hypoperfusion [6]. Age-related changes in the cerebral vasculature predispose the brain to neurovascular injury [6]. Post-stroke dementia often follows large vessel strokes and stenosis, while multi-infarct dementia results from cumulative cortical and arteriolar infarcts. Mixed dementia arises when vascular pathology coexists with neurodegenerative changes, such as those seen in AD [35]. CSVDs primarily affect the small blood vessels in the brain, leading to white matter lesions, lacunar infarcts, and microbleeds [42,43,44]. These vascular abnormalities are closely associated with cognitive decline, and they can disrupt the brain’s ability to function properly. CSVDs can lead to both ischemic and hemorrhagic events in the brain. These events contribute to vascular dysfunction and can result in stroke-like symptoms and cognitive impairment. Degenerative changes in small blood vessels can lead to reduced blood flow, increasing the risk of strokes and VaD.

VCID has a notably high conversion rate to dementia, with 40–60% of individuals developing full-blown dementia within 5 years of diagnosis [38]. This points to its rapid, stepwise progression and potential for significant morbidity and mortality. Early identification and intervention are crucial for mitigating cognitive decline and may allow for the implementation of therapeutic or preventative strategies to slow disease progression.

### 2.4. Interplay Between Vascular Pathology and Neurodegeneration

Aging is a major risk factor for both vascular dysfunction and neurodegenerative pathologies, and both frequently coexist in older individuals [6,10,41].

While the causal direction between vascular dysfunction and neurodegeneration remains unclear, current experimental models and longitudinal studies are increasingly designed to disentangle these complex interactions. For example, transgenic animal models of AD, such as APP/PS1 mice, have been shown to have vascular abnormalities, such as BBB disruption and cerebral hypoperfusion prior to the onset of amyloid plaque formation and cognitive decline [42]. Conversely, models that induce vascular injury through hypertension, chronic hypoperfusion, or microinfarcts can trigger neuroinflammatory and degenerative changes, providing evidence for a vascular-to-neural causal pathway [43,44]. Supporting this, longitudinal assessments in the zQ175DN mouse model have revealed that changes in CBF and vascular function may precede and contribute to neurodegeneration in Huntington’s disease [45]. A comprehensive review has highlighted how chronic cerebral hypoperfusion leads to reduced oxygen and glucose supply to the brain, directly damaging the BBB, which in turn, mediates neurotoxic effects, oxidative stress, inflammation, and further reduction in cerebral perfusion, thereby accelerating neurodegeneration in Alzheimer’s disease [43,46].

Newer models employ multi-hit paradigms, where both vascular insults and neurodegenerative mutations are introduced, allowing researchers to assess how these pathologies may synergize or interact over time. Epidemiological studies have shown that midlife vascular risk factors (e.g., hypertension, diabetes, and atherosclerosis) are predictive of later-life cognitive decline and brain atrophy, supporting a vascular-first hypothesis. For example, a recent Harvard Aging Brain Study utilized exploratory factor analysis to identify independent latent factors, such as systemic vascular risk, white matter injury, and relative CBF, that independently predicted cognitive decline beyond traditional markers like amyloid and tau. These results suggest that systemic vascular risk contributes to cognitive decline beyond current markers of cerebrovascular injury [47]. Similarly, the Enhancing Neuroimaging Genetics through Meta-Analysis (ENIGMA) study, a single-center prospective clinical observational study conducted at Aarhus University Hospital, Denmark, is investigating how capillary dysfunction and extracellular vesicle characteristics predict cognitive outcomes one year after acute ischemic stroke and transient ischemic attack [48].

Mechanistic models have also explored the interactions between vascular dysfunction and neurodegeneration by combining vascular challenges with intracerebral injection of Aβ peptides in transgenic animals. This approach allows the examination of how vascular lesions contribute to cognitive impairment in the context of AD pathology [49]. Zebrafish models have been used to study the genetic underpinnings of cerebrovascular diseases. For example, perturbation of the FOXC1 gene leads to cerebral hemorrhages and affects platelet-derived growth factor signaling, providing insights into the molecular mechanisms linking vascular dysfunction to neurodegeneration [50].

While these studies collectively contribute to a more nuanced understanding of how vascular dysfunction may initiate or exacerbate neurodegenerative processes, others have highlighted evidence for the reverse relationship: early amyloid deposition may itself impair cerebrovascular function, suggesting a neural-to-vascular causative direction [51,52,53,54].

Increased evidence has highlighted that neurodegeneration during aging can exacerbate vascular dysfunction through increased oxidative stress, inflammation, and endothelial damage [55,56]. For example, in AD, vascular dysfunction accelerates disease progression by impairing the clearance of Aβ and tau proteins. Conversely, AD-related neurodegeneration further aggravates vascular dysfunction by promoting oxidative stress and inflammatory responses. Aβ can also accumulate in the walls of cerebral blood vessels, leading to CAA [31], a condition associated with microbleeds and microinfarcts that impair CBF and oxygen delivery.

Additionally, Aβ is directly toxic to vascular cells, contributing to endothelial dysfunction and vascular rarefaction, thereby worsening cerebral perfusion deficits [26].

Aging-related damage to the NVU contributes to vascular and neuronal injury, ultimately leading to cognitive decline [6]. The interplay between vascular dysfunction and AD pathology creates a vicious cycle that accelerates cognitive deterioration.

On the other hand, several interventions originally developed for AD have demonstrated potential benefits for improving vascular function, including Aducanumab, Donepezil, Galantamine, Memantine, and Nilvadipine [57,58,59,60]. While primarily targeting mechanisms like amyloid plaque reduction and neurotransmitter regulation, these treatments have also been associated with improvements in vascular health. Additionally, lifestyle modifications, including exercise and dietary changes, have shown promise in enhancing both AD and vascular outcomes [57]. Notably, clinical trials for VaD often overlap with those targeting functional outcomes in broader cerebrovascular diseases, reflecting shared underlying mechanisms and treatment targets.

Given the bidirectional relationship between vascular health and neurodegeneration, targeting vascular risk factors presents a promising strategy for preventing or delaying cognitive impairment. Further research is needed to understand how age-related changes in vascular and neural tissues interact, particularly in the context of structural and functional alterations that drive disease progression. Aging is a significant risk factor for both degenerative diseases and vascular dysfunction [6,61,62].

## 3. Recent Progression in Molecular Mechanisms and Potential Targets

Despite extensive research, the precise molecular mechanisms underlying VCID remain incompletely understood. Here, we outlined recent advances in the relevant pathways and potential therapeutic targets. These studies are also summarized in Table 1.

### 3.1. Mechanisms Related to BBB Disruption

VCID constitutes many vascular changes that can be detrimental to the body and occur with natural aging. The BBB acts as a division between the central nervous system and peripheral vascularization and is composed of cerebral endothelial cells, astrocytes, pericytes, microglia, and basement membrane [63]. The cellular and non-cellular components of the BBB work in harmony to maintain structural and functional integrity [64]. Disruption in the BBB can cause neurotoxic substances, proteins, cytokines, metabolites, and bacterial products to enter the brain parenchyma, so it is essential in maintaining cerebral homeostasis.

A. Matrix metalloproteinases (MMPs): Alternatively, the BBB permeability can increase due to the possible excess MMP production caused by chronic hypoxia and inflammation induced by age-related vascular changes, which, therefore, contributes to neurodegeneration and functional decline [6]. MMPs have a major role in the breakdown of the extracellular matrix and tight junctions of the BBB, so they have been implicated in VCID due to their effect on BBB permeability, neurovascular unit dysfunction, and microhemorrhages [65].

Recently, MMP 2 and MMP 9 were found to degrade basement membrane proteins and tight junction proteins, which would ultimately be detrimental to the integrity of the BBB as it indirectly manages ion concentrations and passage of major energy components [65]. It has also been found that hypoxia and/or inflammation can cause an increase in the MMP 9 transcript, which could eventually lead to the cleavage of β-dystroglycan between astrocytes and their associated vessel [65]. Experiments also demonstrated that MMP-9 was primarily produced by reactive astrocytes near the BBB during the early stages following SAH (subarachnoid hemorrhage), contributing to decreased BBB integrity [65,66].

B. Glycocalyx: In addition to MMPs, a most recent study showed that brain endothelial glycocalyx is highly dysregulated during aging and neurodegenerative disease, which impairs the BBB in aging and disease. The glycocalyx layer is a structural component of the BBB that is composed of proteoglycans, glycoproteins, and glycolipids. Aging and neurodegenerative disease cause dysregulation of glycocalyx, which was demonstrated by a decline in mucin-type O-glycosylation in these conditions. Therefore, restoring the brain endothelial glycocalyx opens a potential avenue for preventing BBB breakdown in age-related central nervous system (CNS) diseases [67].

### 3.2. Mechanisms Related to Endothelial Dysfunction

A. Endothelial nitric oxide synthase (eNOS): It was identified that inhibition of eNOS in cultured human brain microvascular endothelial cells caused significant upregulation of amyloid precursor proteins and beta-site amyloid precursor proteins cleaving enzyme 1 [27]. It has also been reported that eNOS enhances aerobic glycolysis in mouse cortical astrocytes through activation of hypoxia-inducible factor 1-alpha (HIF-1α)-mediated target genes, contributing to the maintenance of high glycolytic activity in astrocytes [68]. Despite the mechanisms through which endothelial NO loss affects cognitive impairment being unclear, it has been noted that inactivation of eNOS through gene deletion or a high salt diet increases the phosphorylation of tau proteins, a hallmark of Alzheimer’s pathology in neuronal cells [27]. Additionally, it was identified that high salt diet mice that received L-arginine, an NO precursor, had suppressed phosphorylation of tau [27]. The NO donor 2,2′-(hydroxynitrosohydrazino)bis-ethanamine (NOC-18) has been shown to increase the synthesis of collagen type IV in endothelial cells [69], which potentially links to arterial stiffness and the pathogenesis of dementia [70].

B. N-methyl-d-aspartate receptor (NMDAR): Endothelial dysfunction is also a major contributor to VCID, as the cerebrovascular endothelium plays a role in the CBF, BBB, blood coagulation, immune surveillance, and trophic support of neuronal and glial cells [27]. Endothelial cells line the inner lumen of blood vessels and are exposed to blood and a layer of VSMCs [71]. The endothelial cells play a myriad of roles as they release multiple vasoactive mediators that can induce vasorelaxation or vasoconstriction [71]. A recent study showed that endothelial dysfunction during chronic cerebral hypoperfusion may cause increased serum antibodies against NMDAR [63].

NMDAR dysfunction in both neurons and endothelial cells has been implicated in the progression of neurodegenerative diseases such as AD [72]. Unlike neuronal NMDARs, endothelial NMDARs play a crucial role in mediating neurovascular coupling, a process in which increased neuronal activity leads to enhanced cerebral blood flow [73]. Impaired NMDAR signaling in endothelial cells has been shown to disrupt the BBB and compromise neurovascular coupling, contributing to the altered hemodynamic responses frequently observed in neurodegenerative disorders [71,74].

C. Platelet-derived growth factor (PDGF): Pericytes have been found to play a role in CBF and brain capillary perfusion, as decreased pericyte concentration is correlated with disordered neurovascular uncoupling along with hypoxia through reduced oxygen supply to the brain, further resulting in the loss of structural stability [27]. A study found that white matter pericytes were reduced by 35–45% in subjects with PSD (post-stroke dementia), VaD, AD-VaD, and AD compared to healthy aging controls [63]. The key signaling pathway in pericytes is PDGF, whose disruption causes less pericyte recruitment to the vessel, vessel leakage, tortuosity, microaneurysms, and microbleeds [75].

### 3.3. Mechanism Related to CBF

The Rotterdam Study, published in 2005, was one of the first studies to establish a connection between cerebral hypoperfusion and cognitive impairment with risk of dementia by measuring CBF velocity and cognitive decline WMH, which are regarded as the direct manifestation of chronic cerebral hypoperfusion, are negatively correlated with local perfusion and CBF [63]. These white matter lesions are a classic manifestation of VCID and are associated with lower motor function in people with average and low activity [76].

BBB changes alter the CBF in the brain, which holds approximately 12% of the circulation of cardiac output and consumes 25% of glucose and oxygen [39]. Therefore, even small changes in the CBF can have adverse effects. However, there are many pathologic changes that can reduce CBF, which include vascular structural lesions, cerebral hemodynamic alterations, reduced cardiac output, increased blood viscosity, and several other major risk factors [39].

A. Type IV collagen: Cerebral autoregulation is impacted by baroreceptors, which are controlled by aging-related hypertension and increased pulsatility [6]. Moreover, smaller brain volumes, increased volume of abnormal white matter, and microstructural injury have been linked to arterial stiffness measured as carotid femoral pulse wave velocity [27]. Type IV collagen, which assists in mechanical stability and cell-cell interactions, has been associated with genetic variations that lead to increased carotid artery stiffness [70].

B. Nuclear factor erythroid 2 related factor 2 (Nrf2): Oxidative stress can contribute to vascular aging and therefore also contribute to vascular diseases such as stroke, VCID, and coronary heart disease [6]. It is theorized that an increase in reactive oxygen species (ROS) occurs with age and causes an increased risk for vascular diseases by damaging mitochondrial DNA and increasing cerebral vasomotor dysfunction [6]. Aging causes low-grade inflammation that ultimately leads to increased ROS production, and ROS inhibit the activity of eNOS, which plays a role in matching regional CBF with neural activities [61]. Most importantly, ROS inhibit the anti-inflammatory properties of NO, which is essential for vasculature protection as it plays a role in the inhibition of platelet aggregation, inhibition of endothelial apoptosis, and the preservation of endothelial progenitor cells [6]. Nrf2 is a transcription factor that can control the cellular redox status, which therefore can affect neurovascular coupling, hippocampal activity, and blood–brain barrier disruption [77]. It has been demonstrated that Nrf2 expression and activity decrease with age, and improper functioning of Nrf2 causes endothelial cells to lose protection against physiologic ROS production [61].

C. Insulin-like growth factor 1 (IGF-1) Another factor that could play a role in VCID are vascular smooth muscle cells (VSMCs) which can modulate various responses to environmental stressors, such as regulating blood pressure, organ perfusion, and vascular integrity [23] by surrounding the endothelial cell layer and, therefore, regulating cerebrovascular resistance and blood flow to downstream capillary beds [22]. Changes in the renin-angiotensin system (RAS) can be involved in neurodegenerative disease [23]. The RAS system is necessary for vasoconstriction, while NO plays a role in vasodilation [61].

Furthermore, phenotypic switching of the VSCM is critical for healthy brain aging due to its influence on normal vessel formation and vascular stability [19]. When improperly regulated, VSMCs can adopt maladaptive phenotypes, such as macrophage, foam cell, and chondrocyte-like characteristics, in vascular pathologies. In the context of CSVDs, this phenotypic transformation contributes to aging and age-related cognitive impairment [19].

IGF-1 is a protein that potentially targets the phenotypic switching of VSMCs due to its known implication in the worsening of cerebrovascular disease and signs of VCID from human and animal models [23]. It has been identified to decrease with age, likely causing the acceleration of cerebrovascular disease and signs of VCID [23]. It was identified that the deficiency of IGF-1 leads to disordered adaptation of VSMCs to hypertension and neurovascular coupling dysfunction [23].

## 4. Implications in Management

There are currently no cures for VCID and AD. Early detection and management of vascular risk factors (e.g., hypertension, diabetes, and dyslipidemia) can reduce dementia risk. Therapeutic strategies that enhance CBF, protect endothelial function, and reduce vascular inflammation may help mitigate neurodegeneration. This can further aid in focusing on preventative measures and risk targeting with treatments targeting chronic cerebral hypoperfusion and neuroinflammation, simultaneously or individually [65].

### 4.1. Diagnostic Techniques

There are several neuro-imaging techniques that can assist with the diagnosis and treatment of VCID. A neuro-imaging technique that helps with the diagnosis and identification of VCID is diffusion tensor imaging (DTI), which can be used as an indicator for early WM damage caused by systemic vascular injury, as these tracts are disrupted in aging and SVD [78]. WMH volume is also another SVD biomarker that has been used along with a composite vascular white matter score [27]. Furthermore, there are several imaging-based measures that can serve as candidate biomarkers for various pathologies such as WMH, infarcts, microinfarcts, brain barr disruption, resting CBF, and increased cerebrovascular resistance [27].

However, it is essential to also note that specific biomarkers may vary based on the stages of VCID. For example, small vessel dysfunction is a biomarker for at-risk asymptomatic patients, and white matter injury is a biomarker for at-risk patients with symptomatic disease [27]. For example, dynamic contrast-enhanced MRI measures BBB permeability in humans [27]. PET imaging can allow for the assessment of AD pathology with ligands targeting fibrillar Aβ and neurofibrillary tau, despite there being caveats with the PET ligands, which are nonspecific in nature [27]. Screening for VCID may be enhanced with the development of cognitive batteries sensitive and specific to VCID, which could be used to evaluate and differentiate VCID and mixed pathologies from forms of dementia without VCID [27]. Furthermore, it must also be considered that a portion of the population may be biomarker positive and cognitively normal, which prompts the need for a multimodal biomarker assessment [27].

### 4.2. Biomarkers

There has been a surge in the identification of specific biomarkers for more specific and sensitive diagnoses [38]. Blood-based biomarkers continue to be effective indicators for VCID despite the central diagnostic criteria being primarily based on neuro-imaging [38]. A classic blood-based biomarker for BBB dysfunction is MMP [38]. MMPs 2 and 9 have been associated with the degradation of the basement membrane and tight junctions of the BBB [38]. Furthermore, not only a prognostic advantage but also possible therapeutic interventions can be acquired by the identification of pro-inflammatory proteins [38] for those at risk for developing VCID. Alternatively, in some circumstances, peripheral vascular changes or systemic metabolic variations may precede cerebrovascular abnormalities, which may open opportunities to detect early biomarkers of cognitive decline [27]. There are also several potential biomarkers for BBB disruption leakage, and one such example is the soluble PDGFR-β [79]. PDGB-BB binds to PDGFR-β on pericytes and regulates survival, migration, apoptosis, proliferation, and differentiation, which impact pericyte-regulated BBB integrity [79].

Integrative data-driven approaches are also advancing a comprehensive understanding of the multifactorial nature of AD and VCID. One such analysis of late-onset AD revealed that intra-brain vascular dysregulation is an early pathological event in disease progression. This approach integrates multiple biomarkers, including Aβ deposition, metabolic activity, vascular function, and structural brain changes, to characterize disease progression [80].

### 4.3. Therapy

Several studies and clinical trials highlight the link between vascular dysfunction and AD, suggesting that targeting vascular health may be beneficial for managing or preventing AD. Notable examples include trials investigating the effects of antihypertensive medications, such as DIA MIND (Systolic Blood Pressure Intervention Trial Memory and Cognition in Decreased Hypertension) [81] and SMARRT (Systematic Multi-Domain Alzheimer’s Risk Reduction Trial) [82,83,84]. The Dominantly Inherited Alzheimer Network Trials Unit (DIAN-TU) also explores preventive strategies for AD by targeting vascular health [85].

Clinical trials have shown that improving vascular function in AD patients represents a promising therapeutic approach. Treatments such as antihypertensives have demonstrated potential benefits [86]. Recent findings suggest that primary hypertension medications, specifically angiotensin receptor blockers (ARBs) and angiotensin-converting enzyme inhibitors (ACEis), may offer neuroprotective properties. These drugs can cross the BBB, promote vasodilation, enhance CBF, and reduce brain atrophy, as evidenced in the RADAR trial [87,88]. Additionally, vasodilators like cilostazol, a phosphodiesterase-3 inhibitor commonly used to treat peripheral vascular disease, have shown potential to improve cognitive function in patients with VCID and could have similar benefits for AD patients, particularly those with underlying vascular pathology [89].

Several novel combined approaches have been reported for AD treatment [90]. One example involves a bispecific antibody therapy designed to target both Aβ and tau proteins in the brain. This antibody is engineered to bind to transferrin receptors on endothelial cells, enabling it to cross the BBB more efficiently. Using this antibody transport vehicle (ATV), researchers have successfully delivered anti-BACE1 (Beta-site APP Cleaving Enzyme 1) antibodies in APP (amyloid precursor protein) transgenic mice, resulting in reduced Aβ levels and plaque formation [91,92,93].

Combining pharmaceutical and lifestyle interventions that address cardiovascular risk factors presents another promising strategy with the potential for greater efficacy [90]. For instance, exercise combined with antihypertensives and statins appears to be more effective than either intervention alone in mitigating cardiovascular risks such as hypertension and dyslipidemia, both of which are associated with increased AD risk [94].

Recent perspectives on AD treatment emphasize the importance of addressing BBB dysfunction, particularly during the early stages of dementia, before extensive neurodegeneration occurs. Several therapeutic strategies have been proposed, including strengthening tight junctions between endothelial cells to restore BBB integrity, using anti-inflammatory agents to reduce neuroinflammation to mitigate BBB leakage, or targeting specific signaling to promote pericyte survival. Ongoing clinical trials are also exploring the combination of these BBB-targeted strategies with disease-modifying therapies, such as anti-amyloid antibodies, particularly in patients with early-stage Alzheimer’s disease [95,96,97,98].

Additionally, therapeutic strategies targeting inflammation and oxidants have also been applied in VCID treatment. The neurofilament light chain (NfL) protein has emerged as a key biomarker, showing strong positive correlations with cognitive impairment, neurodegeneration, hypoxic brain injury, and cardiac disease, and elevated NfL levels have also been significantly associated with lower cognitive performance [99]. Treatment with PNA5, a novel glycosylated angiotensin 1–7 peptide and Mas receptor agonist, has been shown to reduce inflammation by targeting Mas receptors on neurons, microglia, and vascular endothelial cells, resulting in a significant decrease in NfL levels [99].

TNF-α is another promising therapeutic target due to its role in MMP activation and microglial-induced inflammation [65]. The selective TNF-α inhibitor XPro1595 binds specifically to soluble TNF-α while preserving the immune-supportive functions of transmembrane TNF-α. This compound has been shown to reduce microglial and astrocyte activation in the hemiparkinsonian rat model [65].

Antioxidant therapies are also being explored for VCID. Resveratrol, a natural antioxidant, has demonstrated neuroprotective effects in mouse models of gradual cerebral hypoperfusion. Biweekly administration reduced cognitive deficits and inflammation [100]. Another potential candidate for future therapeutic strategies is Nrf2, which is a master regulator of antioxidant and anti-inflammatory gene expression [77].

Notably, isosorbide mononitrate and cilostazol have shown promise in improving BBB integrity and reducing neuroinflammation. Isosorbide mononitrate promotes NO production, enhancing vasodilation, while cilostazol improves cerebral perfusion and reduces inflammation through similar vasodilatory mechanisms. Both compounds are currently being evaluated in clinical trials (LAC-1 and LAC-2) aimed at preventing recurrent vascular events and improving functional outcomes in patients with prior lacunar strokes, a common presentation of CSVDs [65].

### 4.4. Prevention

Several risk factors for VCID include smoking, hypertension, diabetes, obesity, and hypercholesterolemia [6]. For example, smoking has been associated with impaired verbal learning in women and cardiovascular disease in men, alongside vascular dysfunctions like endothelial impairment and arterial stiffness [101].

Hypertension contributes to microvascular damage and structural brain changes and is also an indirect risk factor through its association with other diseases. Blood pressure control has been shown to reduce dementia risk, with the SPRINT-MIND trial linking intensive BP management to lower rates of cognitive decline impairment [81,83,102]. Obesity, by reducing nitric oxide bioactivity, further exacerbates endothelial dysfunction, which is central to VCID pathology [103]. It was also found by a study of 4.6 years that patients with diabetes had worsening processing speeds along with markers of cerebrovascular and degenerative pathology on MRI [104]. While diabetes increased dementia risk, comorbid hypertension and hyperlipidemia did not further amplify it in a separate cohort study [105].

Elevated cholesterol is also a known contributor to cerebrovascular damage, leading to inflammation, ischemia, and cognitive decline [68]. Doppler studies confirm stronger associations between VCID and atherosclerosis compared to AD [106]. The APOE gene, a lipid transporter linked to dementia risk, is also implicated in VCID, highlighting potential therapeutic targets such as antibodies, antisense oligonucleotides, and gene therapy [106]. Hyperhomocysteinemia, characterized by elevated homocysteine levels, is another contributor to VCID, associated with decreased capillary density and brain tissue damage. Atorvastatin has shown efficacy in reducing plasma homocysteine and improving cognitive performance in models of this condition [78,107].

Sex-based differences also play a role. Although women generally have lower stroke, obesity, and diabetes rates pre-menopause, these risks rise significantly afterward due to increased adiposity and metabolic dysfunction [108]. High-fat diets have been found to affect middle-aged females more than males, elevating visceral fat and pro-inflammatory cytokines such as TNF-α [109]. Ethnic disparities are notable as well. Hispanic and African American populations show higher rates of obesity, diabetes, and hypertension, contributing to greater VCID vulnerability [27]. These findings underscore the importance of considering sociocultural and demographic factors in prevention strategies.

While considering preventative measures, it is also essential to account for lifestyle modifications or improvements that can alleviate the symptoms of VCID or delay its progression. There are possible diet changes and activity changes that can impact symptom management more efficiently. For instance, it was found that intermittent fasting can enhance cognitive ability, neurotropic factor production, synaptic plasticity, and mitochondrial biogenesis while also playing a role in the amelioration of vascular pathology and cognitive impairment in rodent VCI models [38]. As VCID is commonly linked to hypercholesterolemia and vessel stenosis, managing cholesterol levels is critical. High-cholesterol diets have worsened vascular conditions in TGF-β overexpressing mice [65]. A recent study demonstrated that simvastatin, an anti-cholesterol drug, prevented cognitive defects in adult and aged wild-type mice that did not undergo a high-cholesterol diet and adult TGF-beta mice that underwent a high-cholesterol diet by targeting white matter inflammation and Galectin-3 (Gal-3) microglial cells, implicated in Alzheimer’s and upregulated in brain injuries [110,111]. Exercise has shown positive effects; 4 weeks of physical activity improved CBF in mice with carotid artery stenosis [112].

## 5. Limitations and Future Directions

There are several gaps in our current understanding of the complex relationship between the vascular system and degenerative diseases, highlighting areas where further research is needed.

### 5.1. Therapeutic Targets

While we know that vascular dysfunction is a common feature in many degenerative diseases, the precise mechanisms underlying this dysfunction, particularly the interaction between vascular dysfunction and the progression of degeneration, remain incompletely understood. Elucidating the specific molecular and cellular pathways involved in vascular damage and their crosstalk with degenerative disease pathology is crucial for developing targeted therapies. Combining data from various sources, such as genomics, proteomics, and clinical records, can provide a comprehensive view of vascular dysfunction in degenerative diseases. Advanced data integration techniques and artificial intelligence can help uncover hidden patterns and relationships.

### 5.2. Biomarkers for Vascular Dysfunction

Identifying reliable biomarkers that can detect early signs of vascular dysfunction and degenerative diseases is an ongoing challenge. Research efforts are needed to discover and validate biomarkers that can predict the risk of degenerative diseases and monitor vascular health.

### 5.3. Interplay Mechanisms

The interplay between vascular dysfunction and neurodegenerative pathology is not fully understood. The complex relationship between the vascular system and degenerative diseases is an area of ongoing research. A comprehensive understanding of the natural mechanisms for the onset and progression of dementia in the context of vessels and specific tissue is essential for a deeper understanding of vascular dysfunction and degenerative disease. Such an understanding is pivotal for gaining deeper insights into vascular dysfunction and degenerative diseases.

### 5.4. Translational Research

Bridging the gap between basic science and clinical application remains a major challenge in developing effective treatments for degenerative diseases such as VCID. Several limitations persist, particularly in the translation of promising laboratory findings into clinical interventions:

a. Animal Models: Due to the heterogeneity of vascular disorders, no single animal model can fully replicate the complex pathology of VCID. Current models typically reflect only specific aspects, such as hypoperfusion, ischemia, hypertension, or hyperhomocysteinemia [27]. For example, bilateral carotid artery ligation (BCAL) in rats or mice induces white matter damage, increased BBB permeability, and cognitive impairment [49]. Bilateral common carotid artery stenosis (BCAS) reduces CBF and increases BBB permeability in white matter [44]. Asymmetrical carotid artery stenosis (ACAS) produces microglial activation and neuronal loss [44]. Vessel occlusion (2-VO, 3-VO, and 4-VO) models in rats cause varying degrees of cerebral hypoperfusion and cognitive dysfunction, with 4-VO showing extensive brain damage [44]. However, these models do not fully mimic the white matter lesions seen in human small vessel disease (SVD), which primarily result from chronic vessel wall degeneration rather than global ischemia or embolic events [27].

Other relevant models include spontaneously hypertensive stroke-prone (SHRSP) rats, which develop stroke lesions, sometimes accompanied by white matter changes depending on preconditions [49]. Pdgfr^(−/−)^ mice and APOE genotype models, which are used to study BBB dysfunction and AD-related pathologies [49]. CADASIL and CARASIL models, which represent monogenic forms of CSVDs and early-onset VCID. Notch3 mutation models mimic progressive white matter damage and aging-related pathology [44,113]. CAA models show amyloidosis, neuroinflammation, and hemorrhages [44]. Type 2 diabetes models also illustrate vascular abnormalities linked to dementia [44]. Despite several advantages, some models pose practical challenges; for example, the CCA stenosis model in mice requires high surgical precision due to the fragility and small size (<0.66 mm diameter) of mouse arteries [114].

In addition, significant physiological differences between animal models and humans, such as lifespan, white matter composition, arterial morphology, Aβ clearance, APOE isoforms, lipoprotein profiles, and immune responses, limit the translational relevance of these studies [49]. Most models do not capture the mixed, progressive, and late-onset nature of VCID observed in patients [27]. Larger animal models may offer advantages due to greater white matter volume and more human-like arterial structures [49].

b. Long-Term Outcomes for Aging Studies: Many studies focus on short-term outcomes of vascular interventions. There is a need for long-term follow-up to assess the durability and sustained benefits of vascular-focused therapies in degenerative diseases.

## 6. Conclusions

Emerging evidence suggests that age-related vascular dysfunction plays a critical role in the pathogenesis and progression of neurodegenerative diseases. As summarized in Figure 1, while aging-induced vascular pathologies are known to contribute to neurodegeneration, several new insights have been reported. These include impaired vascular endothelial function due to dysregulation of eNOS, NMDA receptors, and PDGF; reduced CBF associated with irregularities in collagen IV, Nrf2, and IGF-1; and disruption of the BBB mediated by MMPs and the glycocalyx. These findings challenge traditional views, positioning vascular dysfunction as an early pathogenic trigger. Consequently, targeting vascular pathologies may offer a promising strategy to slow dementia progression and potentially prevent AD.

Conversely, aging-related neurodegeneration can exacerbate vascular dysfunction through oxidative stress, inflammation, and deposition of neurotoxic proteins such as Aβ and tau within vascular walls. This bidirectional interaction forms a vicious cycle that accelerates cognitive decline, highlighting the complex interplay between vascular and neural factors in dementia. Integrating vascular assessments and interventions into clinical practice is therefore essential. We hope this review inspires continued research and supports the development of precision medicine approaches that incorporate vascular health in the management of neurodegenerative diseases.

**Table 1 biomolecules-15-00712-t001:** Summary of mechanisms of VCID.

Mechanism	Study Model	Main Findings	References
Chronic cerebral hypoperfusion	Rodents and humans	- Rotterdam Study, published in 2005, was one of the first studies to establish a connection between cerebral hypoperfusion and cognitive impairment with risk of dementia by measuring cerebral blood flow velocity and cognitive decline - White matter hyperintensities are a direct manifestation of chronic cerebral hypoperfusion and are negatively correlated with local perfusion and cerebral blood flow - CCH also plays a role in Aβ deposition in the cerebrovascular area and brain parenchyma - Reduced levels of MAG:PLP1 and increased VEGF (vascular endothelial growth factor) in postmortem brain tissue in VaD and AD	[6,39,63,65,115]
White Matter Changes	Human	- White matter lesions are a classic manifestation of VCID and are associated with lower motor function in people with average and low activity - Increased total and periventricular white matter hyperintensity burden associated with decreased gait performance over time - White matter hyperintensity volume is also a SVD biomarker	[27,63,76,116]
Blood–brain barrier disruptions		- Blood–brain barrier disturbances tend to be found early in chronic cerebral hypoperfusion, contributing to the deterioration of white matter and the development of cognitive impairment - BBB permeability can increase due to the possible excess production of matrix metalloproteinases due to chronic hypoxia and inflammation - Matrix metalloproteinases have a major role in the breakdown of the extracellular matrix and tight junctions of the blood–brain barrier	[6,63,65,77]
Inflammation	Rodent	- Hyperhomocysteinemia diet induced vascular inflammation, microhemorrhages, and cognitive decline - Elevated serum cholesterol in atherosclerosis can advance the onset of inflammation - Conditions such as atrial fibrillation and sleep apnea can increase systemic inflammation	[65,106,117,118]
Increased Oxidative stress		- Reactive oxygen species inhibit anti-inflammatory nitric oxide, which is essential for vasculature protection - Improper functioning of NRF2 causes endothelial cells to lose protection against physiologic ROS production - Reactive oxygen species inhibit anti-inflammatory nitric oxide, which is essential for vasculature protection as it plays a role in the inhibition of platelet aggregation, inhibition of endothelial apoptosis, and the preservation of endothelial progenitor cells	[6,61]
Endothelial dysfunction		- Endothelial dysfunction during chronic cerebral hypoperfusion may cause increased serum antibodies against the N-methyl-d-aspartate (NMDA) receptor - Inhibition of eNOS in cultured human brain microvascular endothelial cells caused significant upregulation of amyloid precursor proteins and beta-site amyloid precursor protein cleaving enzyme 1 - Decreased pericyte concentration is correlated with disordered neurovascular uncoupling, along with hypoxia, through reduced oxygen supply to the brain, further resulting in the loss of structural stability - Impaired hemodynamic response in neurodegenerative diseases like AD can also be caused by defective endothelial NMDAR signaling	[27,63,71]
Cerebral small vessel disease (CSVD)	Human	- Affects arterioles, capillaries, and venules with major pathologies, such as arteriosclerosis and cerebral amyloid angiopathy - Features of small vessel disease include white matter hyperintensities, recent small subcortical infarcts, lacunes of presumed vascular origin, enlarged perivascular space, microbleeds, and brain atrophy	[35]

## Figures and Tables

**Figure 1 biomolecules-15-00712-f001:**
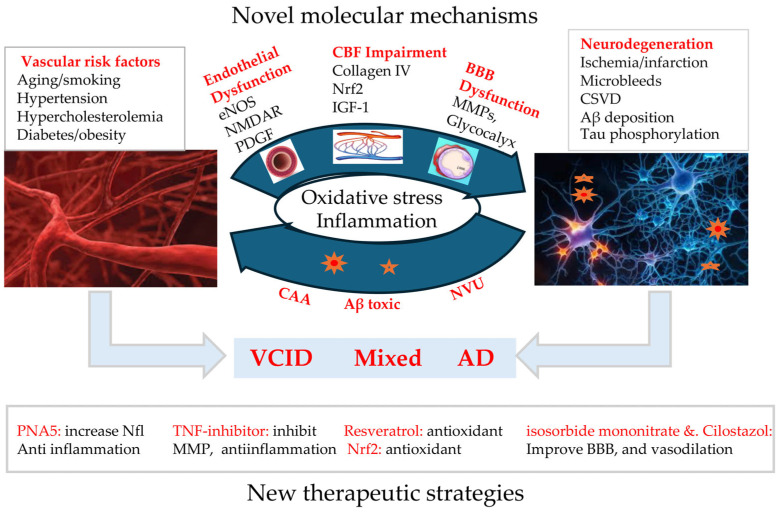
Overview of new insights discussed in this review. This figure summarizes recent discoveries in molecular mechanisms and the development of novel therapeutic strategies targeting vascular pathology. CBF: cerebral blood flow; BBB: blood–brain barrier; VCID: vascular contributions to cognitive impairment and dementia; AD: Alzheimer’s disease; CAA: cerebral amyloid angiopathy; NVU: neurovascular unit; CSVDs: cerebral small vessel diseases; NMDAR: N-methyl-d-aspartate receptor; PDGF: platelet-derived growth factor; Nrf2: nuclear factor erythroid 2 related factor 2; IGF-1: insulin-like growth factor 1; MMPs: matrix metalloproteinases.

## Data Availability

Not applicable.

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
