# Peer review of "Interplay Between Vascular Dysfunction and Neurodegenerative Pathology: New Insights into Molecular Mechanisms and Management"

_biomolecules, 2025, doi:10.3390/biom15050712_

Round 1

Reviewer 1 Report

Comments and Suggestions for Authors

This review presents a comprehensive and timely update of current knowledge on the interplay between vascular dysfunction and neurodegenerative disorders, particularly Alzheimer's disease (AD) and vascular contributions to cognitive impairment and dementia (VCID). The manuscript effectively challenges the traditionally neuron-centric view of AD by emphasizing the significant role of vascular pathology as both a contributing factor and a potential early driver of disease progression.

The authors discussed the bidirectional relationship between vascular and neurodegenerative processes, which is particularly compelling. They highlight how age-related vascular dysfunction may initiate or exacerbate neuronal damage, while neurodegeneration may, in turn, impair vascular health, creating a vicious cycle that accelerates cognitive decline. This integrated perspective offers valuable direction for future research and therapeutic development.

Overall, the manuscript is well-structured, insightful, and makes a strong contribution to the growing field of neurovascular research.  The diagram illustrating the bidirectional relationship between vascular dysfunction and neurodegeneration enhances reader comprehension.

Some minor suggestions are provided to further strengthen the manuscript:  

1.     Clarify the cause-effect relationship: While the review mentions that the causal direction between vascular dysfunction and neurodegeneration remains unclear, a more detailed discussion of how current experimental models or longitudinal studies are working to untangle these interactions would provide valuable context.

2.     Highlight dual-benefit interventions: While the potential of targeting vascular dysfunction is noted, including examples of emerging interventions or clinical trials originally designed to treat AD but also shown to improve vascular function would add practical and translational depth to the review.

Reviewer 2 Report

Comments and Suggestions for Authors

This review article effectively highlights the critical role of vascular health in the pathogenesis and progression of neurodegenerative diseases. The authors emphasize the importance of integrating vascular insights into our understanding of cognitive decline and underscore the potential for innovative therapeutic strategies targeting the vascular system.

Overall, the manuscript highlights the value of vascular-focused research in reshaping our approach to neurodegenerative diseases, offering important insights for both clinical and research audiences.

This reviewer offers the following suggestions to enhance the clarity and impact of the manuscript.

  1. To strengthen evidence-based support: Referencing key studies or clinical trials that demonstrate the benefits of targeting vascular dysfunction would reinforce the review’s central argument and enhance its scientific rigor.
  2. To expand on emerging strategies: Including specific examples of the emerging strategies mentioned, such as pharmacological agents, lifestyle interventions, or biomarker-based approaches, would provide readers with a clearer understanding of current progress in the field.

Reviewer 3 Report

Comments and Suggestions for Authors

This review article provided an overview on the relationship between vascular dysfunction and neurodegeneration related to dementia and/or Alzheimer Disease. It is overall well structured and comprehensively written and provided a wide spectrum of information in the field. The following points are recommended to improve the manuscript's readability. 

  1. There are a lot of abbreviations. A list of abbreviations would be useful for readers.
  2. The authors may include more detailed information on the perspectives of therapy that targets BBB disruption for dementia. 
  3. The following places are too generally written and require specification: (a) lines 245 to 248; (b) lines 261 to 262: "Furthermore, the glycolytic function of astrocytes can be further impaired by microglia stimulated by endothelial NO deficiency"; (c) Same for the sentence "NO has also been implicated in increased collagen type IV synthesis, hence also leading to increased arterial stiffness". Do you mean a decreased endothelial NO? (d) lines 278-279: what are the functions of endothelial NMDAR? (e) line 332: please specify the phenotypic switch of the VSMC. 

Round 2

Reviewer 1 Report

Comments and Suggestions for Authors

The authors have addressed my previous comments. No further comments.

Reviewer 2 Report

Comments and Suggestions for Authors

NO more comments.

Reviewer 3 Report

Comments and Suggestions for Authors

All my comments are adequately addressed.